



# 1 Combining temperature rate and level perspectives in emission
# 2 metrics

Borgar Aamaas[1], Terje K. Berntsen[1,2], Jan S. Fuglestvedt[1], Glen P. Peters[1]
[1]CICERO Center for International Climate Research, PB 1129 Blindern, 0318 Oslo, Norway
[2]Department of Geosciences, University of Oslo, Norway
*Correspondence to*: Borgar Aamaas (borgar.aamaas@cicero.oslo.no)
**Abstract.** The ultimate goal of the United Framework Convention on Climate Change, which is reconfirmed by the
Paris Agreement, is to stabilize the climate change at level that prevents dangerous anthropogenic interference, and it
should be achieved within a time frame that allow the natural systems to adapt. Numerous emission metrics have been
developed and applied in relation to the first target, while very few metrics have focused on the second target regarding
rate of change. We present here a simple and analytical physical emission metric based on the rate of global
temperature change and link that to a metric based on a target for the temperature level. The rate of change perspective
either can supplement the level target or can be considered together in one commitment that needs one combined
metric. Both emission metrics depend on assumptions on a temperature baseline scenario. We give some illustrations
on how this framework can be used, such as different temperature rate and level constraints based on the
Representative Concentration Pathways. The selection of the time horizon, for what time period and length the rate
constraint is binding, and how to weight the rate and level metrics are discussed. For a combined metric, the values
for short-lived climate forcers are larger in periods where the critical rate is binding, with larger temporal increases
during the rate constraint period as the atmospheric perturbation timescale of the species becomes shorter. Global $CO_2$
emissions remain the most important, or among the most important, drivers of temperature rates even during periods
of binding rate constraints.

## 22    1    Introduction
Human activity causes emissions of a range of gases and particles that alter the climate (Myhre et al., 2013), which
has wide reaching consequences (IPCC, 2014). Article 2 in the United Framework Convention on Climate Change
(UNFCCC) states that the ultimate objective is "stabilization of greenhouse gas concentrations in the atmosphere at a
level that would prevent dangerous anthropogenic interference with the climate system. Such a level should be
achieved within a time frame sufficient to allow ecosystems to adapt naturally to climate change, to ensure that food
production is not threatened and to enable economic development to proceed in a sustainable manner" (UNFCCC,
1992). This statement has two specific goals. The first is a long-term stabilization of the climate, e.g. below 1.5 or 2
$^0$C as in the Paris Agreement (UNFCCC, 2015). Less attention has been given to the second target, which concerns
the rate of climate change that allows ecosystems to adapt.



According to Diffenbaugh and Field (2013); LoPresti et al. (2015); Settele et al. (2014) , many plants and animals will
not be able to keep track with climate change in the 21st century in the mid- and high-range climate change scenarios,
the Representative Concentration Pathways RCP4.5, RCP6.0, and RCP8.5. The quicker the temperature increase, the
larger is the velocity of climate change, which is a measure of how fast temperature isotherms are moving towards the
poles (Loarie et al., 2009). A temperature increase of 0.2 $^{0}$C/decade over the 21st century will result in a larger climate
velocity than the dispersal capacity for a number of plant and animal species, and even 0.1 $^{0}$C/decade is critical for
some. The global temperature has historically varied by as much as 0.2 $^{0}$C/decade for single decades (Hansen et al.,
2010; Morice et al., 2012; Smith et al., 2008), while the literature suggest that such trends are critical for plants and
animals when lasting several decades. The required movement of plants and animals will be the fastest in extensive
flat landscapes. These studies show there might be some maximum temperature rate that is acceptable, just as an
absolute global temperature level of increase of e.g. 1.5 or 2 $^{0}$C is seen as tolerable.
If only considering $CO_2$, the timing and magnitude of the largest temperature rate increase is determined by the $CO_2$
emissions peak and level (Bowerman et al., 2011). This perspective was widened by Matthews et al. (2012), that found
that the rate of warming depends linearly with the rate of increase of $CO_2$ cumulative emissions, while the eventual
warming level depends on the total cumulative $CO_2$ emissions. For a cumulative total of $CO_2$ emissions, the
temperature rate is highly dependent on the pathway of emissions (LoPresti et al., 2015). O'Neill et al. (2006) propose
interim targets, as opposed to near- or long-term targets, such as a 2050 target of about 430 ppm $CO_2$- equivalents that
ensure rate warming of no more than 0.1 $^{0}$C/decade, or 550 ppm $CO_2$- equivalents for 0.2 $^{0}$C/decade. Kallbekken et
al. (2009) argue for limiting the cumulative $CO_2$ emission budget in the next few decades to limit the rate of warming,
while such a rate approach enables short-lived climate forcers (SLCFs) also to be included in climate policy.
UNFCCC (1992) states in Article 3 that that climate policies should be "cost-effective so as to ensure global benefits
at the lowest possible cost" as well as "be comprehensive", and "cover all relevant sources, sinks and reservoirs of
greenhouse gases." In order to operationalize Article 3, emissions of various species must be made comparable, which
can be done by applying emission metrics. The most widely used metric is based on the cumulative global mean
warming effect over a time horizon (i.e., the Global Warming Potential (GWP), (IPCC, 1990)) or the global
temperature increase at some point in the future (e.g. the Global Temperature change Potential (GTP) (Shine et al.,
2007; Shine et al., 2005)) as a measure of dangerous anthropogenic interference. The GWP is not directly linked to
the rate or level targets, while the GTP with a time horizon is compatible with the level target, such as limiting global
temperature increase to less than 2 $^{0}$C.
Here, we take Article 2 of the UNFCCC as our starting point, assuming that climate policies will be targeted to keep
both the rate of change as well as the long-term stabilization below certain thresholds. To make the GTP-metric
suitable for the long-term stabilization target, as discussed above, a key point is how to set the time horizon. Following
Tol et al. (2012), the time horizon should be set to when the temperature stabilization target is expected to be reached
(binding). This requires an a priori assumption about the future climate development and we will denote this the
*baseline scenario*.



Based on Article 2 of the UNFCCC, there is a need for a transparent metric compatible with the rate target. Such a
metric concept could be explored due to its potential usefulness, as well as providing insight, even though the political
feasibility might be low. As the temporal development in temperature is also linked to the rate target, GTP in some
form is also relevant for the rate metric. Following the same argument as for the GTP/stabilization target above, we
assume that with the climate scenario used there is some period where the rate of change will be above an acceptable
threshold. If not, then the rate of change is considered not to be a problem and policies should only focus on the long-
term stabilization target. With this framework, we suggest a modified GTP metric to address the rate target by
quantifying how pulse emissions at a given time contribute to warming within the period where the rate constraint is
binding.
Most of the developed metrics related to the rate goal are not purely physical, but also include economics. Manne and
Richels (2001) presented an emission metric that in principle is the Global Cost Potential (GCP) based on an economic
model that considered both the absolute temperature change and rate of temperature change, which was revisited by
Ekholm et al. (2013). Wallis and Lucas (1994) reformulated global warming potentials to include rate of change, and
Peck and Teisberg (1994) discussed optimal carbon emission trajectories given costs of different warming rates and
levels. These metrics are based on economic modeling, which is not necessary transparent to the metric user.
Kirschbaum (2014) developed the physical metric climate-change impact potential (CCIP) by giving identical weight
to three parameters, the temperature increase, rate of warming, and accumulated warming. He focused on emission
cases over a time period of 100 years until 2100 and did not develop a general rate metric which for instance separate
between periods that are rate constraint binding and not binding.
The rate of change perspective can be adopted to supplement a long-term stabilization target in two different ways.
One may regard the rate of warming and the long-term stabilization as two independent environmental issues (although
affected by the same emissions, much like air quality and climate change) or it may be regarded as one single problem.
In the former case, separate commitments in terms of weighted emissions (i.e. two sets of $CO_2$-equivalents and
emission metrics) for each of the problems could be negotiated, while with the latter approach a single commitment
is given using one common emission metric that includes a weighting to account for both the rate of change and the
long-term warming. The different configurations should all be based on the same basket of gases and particles, as all
the species affect both the level and rate of change.
In Sect. 2, we show alternative potential rate metrics. We present our suggested analytical and simple emission metric
that is relevant for the objectives addressing both the long-term stabilization and the rate of change in Sect. 3. In Sect.
4, we give some applications of these metrics for the 21st century. We discuss the implementation in a mitigation
policy framework, applications of the developed metrics and the linkage to cost-effectiveness in Sect. 5. Section 6
concludes.





## 2 Alternative rate metrics

Given that the rate of change causes damage (according to Article 2), one would intuitively attempt to derive a metric based on the rate of change (i.e. based on $R(t) = \frac{d(\Delta T(t))}{dt}$). To illustrate why we would argue that the framework described above (using a modified GTP) is a better approach, a schematic temperature response after a pulse emission is shown in Fig. 1. The temperature rate of change is always largest at the time of emissions ($t_e = t_{Rmax}$) and gradually reduced until $t_{R0}$, where the temperature rate turns negative. On the other hand, the absolute temperature change is positive throughout the period, with the largest increase at $t_{R0}$.

Possible metrics based on R(t)

1. $AM(t_e, t) = R_{max}(t_e, t)$

   Choosing the time horizon so that the rate of change is at its maximum, is equal to setting the time horizon effectively to zero. Due to the atmospheric decay given by an impulse response function, the rate of temperature increase is always largest as $t_e \rightarrow 0$, as seen in Fig. 1 in the Supporting Information. Then the relative metric would be equal to the ratio of radiative effects, that is a similar framework as applied in the Radiative Forcing Index (RFI) (IPCC, 1999) , which has been criticized (e.g., Fuglestvedt et al., 2010).

2. $AM(t_e, t) = \int_{t_e}^{t_{R0}} R(t_e, t') dt'$

   With this approach, we integrate the rate only when the rate of change is positive. However, this integral is equal to the $AGTP(t_e, t_{R0})$. Thus, a relative metric would use a (potentially very) different time horizon for different species (see also Sect. 2 in the Supporting Information). For very SLCFs like BC, $t_{Ro}$ would be less than a year after emissions.

3. $AM(t_e, t) = \int_{t_e}^{t_L} R(t_e, t') dt'$

   where $t_L$ is the time horizon for the long-term stabilization target. This definition is exactly equal to the proposed AGTP for use in relation to the long-term stabilization target.

## 3 Definition of rate and level metrics used in this paper

Based on the general approach described in the introduction, we develop the formal definition of our rate and level metrics. The metrics are defined for pulse emissions (see Fig. 1), noting that metrics for a sustained emission change or any future emission path can easily be derived from the pulse metrics (Fuglestvedt et al., 2010). For both metrics, we assume that a general binding target constraint occurs over some time period. As the calculation of these metrics depends on the baseline scenario for the temperature development, the dependence on scenarios are first shown.



### 2.1 Dependence on scenario

Both the rate and level metrics depend on the choice of the baseline scenario. There is a critical temperature rate and temperature level, which gives binding constraints for the metrics. The baseline scenario determines the time horizon or range of time horizons depending on when the targets are binding. This is illustrated in Fig. 2, where the level constraint is binding between $t_{L1}$ and $t_{L2}$, while the rate constraint is binding between $t_{R1}$ and $t_{R2}$. The time horizons for the two targets will of course be different, with the time horizon for the rate metric always shorter than for the level metric. The choice of a proper baseline scenario is not straightforward and beyond the scope of this paper. In Sect. 8 in the Supporting Information, we combine different Representative Concentration Pathways (RCPs) with different rate and level constraints, and show how this determines possible time horizons.

### 2.2 Level metric

The time-dependent GTP is a temperature emission metric that is well-known to be used relative to the level target, for instance the 2 $^0$C target (Shine et al., 2007). The time horizon can potentially be set to when 2 $^0$C global warming is reached (Joshi et al., 2011), while our framework focuses on the timing of the temperature stabilization. We define here the level metric $AM_L$ for emissions at time $t_e$ for species $i$ given by the level term calculated from the Absolute Global Temperature change Potential (AGTP):

$$AM_L(t_e) = \int_{t'=t_{L_1}}^{t_{L_2}} \Delta T_i(t_e, t') dt' = \int_{t'=t_{L_1}}^{t_{L_2}} AGTP_i(t' - t_e) dt' \qquad (1)$$

The level target period is binding for some limited period between $t_{L1}$ and $t_{L2}$ when the temperature increase is above the target level $\Delta T=T$, for instance T=2 $^0$C, as illustrated in Fig. 2(A). In the case for emission occurring after $t_{L1}$ ($t_e>t_{L1}$), we set t'= $t_e$. The target period could be shortened to one specific year, e.g. year 2100, which would give a metric identical to a time-dependent AGTP for a pulse emission. However, we integrate over a period as the temperature stabilization may in reality occur over a longer period or may not be able to be specified to a single year due to uncertainty in emissions and climate response (Shine et al., 2007). Additional warming above the level is given equal weight throughout the period of binding level constraint. As emissions at time $t_e$ approaches the level target period starting at time $t_{L1}$, the time horizon for the AGTP calculations is gradually reduced.

### 2.3 Rate metric

We propose a physical and analytical rate metric that depends on when the rate constraint is binding. A specific total allowable rate of change for the global temperature over time must be selected that reflects a level of ecological risks that can be tolerated. Natural variability will come on top of the rate of change imposed by anthropogenic forcing. The maximum anthropogenic rate, which then determines the constraint for the metric, is then the difference between the total allowable rate and possible contribution by natural variability. We acknowledge that determining the specific maximum anthropogenic rate is not straightforward; however, the scope of this paper is to lay out the framework for how rate considerations could be implemented in a comprehensive approach following Article 2 of the UNFCCC.

Given the baseline scenario selected, a period when the rate constraint is binding is determined. This is illustrated in Fig. 2(B), where the rate is estimated to be above the threshold, and thus binding, between $t_{R1}$ and $t_{R2}$. The absolute metric value ($AM_R$) for a unit pulse emission at time $t_e$ is defined to be the integral of the temperature change within





this period cf. Eq. (2). A temperature increase is given the same weight whenever in that period this warming occurs,
as any additional warming is equally critical throughout the period of the binding rate constraint.
$$AM_R(t_e) = \int_{t'=t_{R_1}}^{t_{R_2}} \Delta T_i(t_e, t')dt' = \int_{t'=t_{R_1}}^{t_{R_2}} AGTP_i(t' - t_e)dt' \qquad (2)$$
In the case for emission occurring after $t_{R1}$ ($t_{R1} < t_e < t_{R2}$), then ($AM_R$) is given by
$$AM_R(t_e) = \int_{t'=t_e}^{t_{R_2}} \Delta T_i(t_e, t')dt' = \int_{t'=t_e}^{t_{R_2}} AGTP_i(t' - t_e)dt'. \qquad (3)$$
The potential pathways to temperature $T_L$ are many and the dotted line in Fig. 2(B) shows an alternative pathway with
a temperature rate increase at $t_{R1} < t' < t_{R2}$ just below the rate threshold and, thus, gives no binding rate constraint, which
gives $AM_R(t)=0$ for all time horizons. Hence, we consider the damage during the period with a rate constraint
($t_{R1} < t' < t_{R2}$), but not after ($t' > t_{R2}$).
The proposed rate metric can be seen as a special case of the integrated AGTP and identical to the integrated AGTP
within the time window $t_{R1} < t' < t_{R2}$ (Azar and Johansson, 2012; Peters et al., 2011); however, the metrics are different
due to different choices of integration periods. If the constraint period converges to 0, then the metric is identical to
the time-dependent AGTP.
Both the level and rate metric can be normalized to a reference gas e.g. $CO_2$ to form unitless relative metrics $M_L$ and
$M_R$, respectively.
**2.4 Combining level and rate metric**
The rate of change can supplement the long-term target by regarding them as separate environmental issues both
related to climate change, with two separate mitigation commitments and corresponding metrics for the level and rate
perspectives. For the rate issue, the $CO_2$-eq. emissions will be calculated using the rate metric described above, while
for the long-term e.g. the GTP(t) will be used. In this framework, the parties to an agreement would negotiate separate
quantitative emission reductions for a rate agreement and a level agreement, thus effectively weight the importance of
the rate versus the level constraint. It is important to note that emission reductions of any warming species should be
accounted for both commitments. E.g. is a stakeholder reduces their methane emissions, they will get credit for that
under both the rate and level commitments, albeit with different metric and $CO_2$-eq. values.
If the targets rate of change and the long-term warming are regarded as one single coupled problem, then a common
metric needs to be established. However, successfully achieving the combined emission reduction target based on one
combined metric does not automatically assure that both individual targets are reached.
If the two targets are combined into one, the importance of the individual targets must somehow be weighted relative
to each other. This weighting is not a scientific question, but rather involves value judgments and possibly economics,
and as such would be determined through a negotiation process. Here, we illustrate a simple linear weighting by giving
the rate metric a weight $\alpha$, and the level metric $1-\alpha$, where $0 \le \alpha \le 1$. The combined metric for species $i$ normalized to
$CO_2$ ($M_{R\&L}$) is then



$$M_{R\&L,i}(t_e) = \alpha \frac{AM_{R,i}(t_e)}{AM_{R,CO_2}(t_e)} + (1-\alpha) \frac{AM_{L,i}(t_e)}{AM_{L,CO_2}(t_e)}. \tag{4}$$
This metric is a pure level metric when $\alpha=0$ and pure rate metric when $\alpha=1$. For emissions taking place after the rate
constraint is binding, i.e. for $t_e > t_{R2}$, we set $\alpha=0$. Then, only the level target is relevant and level metric values are
applied without any weighting. Thus, if the rate target is met, a combined metric focuses purely on the level target.
Both metrics are integrated over some constraint period, but can also be used for individual constraint years. Since all
the metrics discussed here are defined relative to a baseline scenario, the specific metric values are all known as a
function of (future) time of emissions ($t_e$). This time dependence must be communicated to the stakeholders so they
would know how to make investments that effect emissions over some future time period.
In this framework, the relative weight of the rate constraint versus the long-term level is determined through the policy
choice of $\alpha$, and a single $CO_2$-eq. mitigation commitment is negotiated. As emitters are free to choose which species
to abate, the outcome of the mitigation efforts are more uncertain when applying a common metric (Daniel et al., 2012;
Fuglestvedt et al., 2000).
**3   Results**
We calculate the level metric, rate metric (i.e., setting $\alpha=0$ and $\alpha=1$, respectively), and combined metric values ($M_{R\&L}$)
for the SLCFs $CH_4$ and BC and the long lived greenhouse gas (LLGHG) $N_2O$ based on radiative efficiencies and
perturbation lifetimes from IPCC AR5 (Myhre et al., 2013). Further, the Impulse Response Function (IRF) for $CO_2$
applied is based on the Bern Carbon Cycle Model (Joos et al., 2013), while the IRF for temperature comes from the
Hadley CM3 climate model (Boucher and Reddy, 2008). The schematic temperature response due to pulse emissions
of these species that is compatible with Fig. 2 is given in Fig. S1(B) in the Supporting Information.
Due to the quick response of SLCFs, mitigation of the warming SLCFs has the potential to regulate the temperature
development and, thus, the temperature rate, for short time horizons (see Sect. 2 in the Supporting Information).
Emission reduction of $CO_2$ will also reduce the short-term temperature rate, in addition to reduced long-term warming.
To demonstrate how the metrics presented in Sect. 3 are applied, we have to choose time horizons for binding rate and
level targets. Our default baseline scenario is an illustration of the framework that is not deduced from a specific
scenario. The default case is based on binding rate constraint for the 2031-2050 period and a level target reached in
the 2081-2100 period. All figures in the paper use these constraints unless otherwise explained. We will in Sect. 5.44.4
relate this hypothetical baseline scenario with potential temperature developments. The later part of Sect. 4 focuses
on $CH_4$ and presents different dimensions or choices for these metrics.
**3.1 Different weighting factors**
In Fig. 3, we show metric values for $CH_4$, BC, and $N_2O$ based solely on the rate or level targets, as well as for an equal
weighting ($\alpha=0.5$). Figs. 3(A), 3(C), and 3(E) are the default cases with a 20 years binding period for both rate (2031-
2050) and level (2081-2100), while Figs. 3(B), 3(D), and 3(F) shows how this changes as the binding period is reduced
towards a minimum of 1 year, that is a rate target in 2050 and level target in 2100. The years on the x-axis correspond
to the time of emissions, $t_e$ in Eqs. (2) and (3). As the rate target is no longer binding after 2050, values for a pure rate



metric ($\alpha$=1) are only given before that. Metric values for other weightings ($\alpha$) are shown in Sect. 5 in the Supporting
Information.
For SLCFs, the metric values increase for emissions occurring towards the start of the rate-binding and level-binding
periods, with a second increase towards the end of the rate-binding and level-binding periods for the species with the
shortest perturbation timescales, such as BC. This fluctuating behavior in metric values for $CH_4$ is similar to the
findings of Manne and Richels (2001) with similar rate-binding and level-binding constraints. The metric values for
SLCFs are, for a period, larger when the binding periods start earlier, since a binding constraint in a given year results
in larger metric values than no binding for the SLCFs. One example is the elevated metric values from 2081 and
onwards for BC in Fig. 3(C) compared with Fig. 3(D). However, the differences between a long binding period (Figs.
3(A), 3(C), and 3(E)) and a short one (Figs. 3(B), 3(D), and 3(F)) is generally small. The longer the long binding
period is, the larger the difference. The opposite occurs for LLGHGs as those species have a relatively decreasing
impact on the temperature as the time horizon decreases, thus, they have a relatively smaller role for the short-term
rate change. The higher the $\alpha$ value is set, the larger influence has the rate metric, and the larger variability with time
is in the combined metric value. The deviation of this combined metric value compared to GWP(100) is largest for
BC due to its short perturbation timescale, and the difference is largest for emissions during the rate-binding and level-
binding periods.

### 3.2  Different rate constraints

Depending on the choice of baseline scenario and the rate and level constraints, the timing and length of the periods
when the constraints will be binding will vary.  Next, we consider how the combined metric for $CH_4$ varies depending
on how long the rate constraint period lasts (Fig. 4(A)) and when the rate constraint becomes binding (Fig. 4(B)). The
earlier the rate constraint becomes binding, the larger is the metric value in early 21$^{st}$ century (see Fig. 4(A)). However,
as the rate metric is an integral over the binding rate constraint period (with equal weighting over time), the metric
values during the first part of these periods are in general lower for longer binding periods. On the other side, the peak
at the end is relatively higher for longer binding periods, as the level metric contributes more if we assume no change
of the level target. If both the rate and level targets are moved correspondingly in time, the metric value curves are
identical, just moved. The reduced metric values at the beginning for longer binding periods occur since the
temperature response of $CH_4$ decays at the end of the period, while $CO_2$ give a much longer lasting response. Moving
the rate constraint period without changing the length of the period just moves the metric curve (see Fig. 4(B)).

### 3.3  Different time horizons for the level target

If we only consider the level target, the level metric value at a fixed year is larger the earlier this level target is reached.
Shine et al. (2007) have previously shown this relationship with a pure GTP metric. Fig. 4(C) shows the combined
metric as we move the level target period gradually from 2081-2100 to 2041-2060. As the rate constraint is kept
constant in this illustration, a drop in the combined metric value is observed for all cases at the end of the rate constraint
period (2050). The size of this drop decreases with decreasing distance between the rate-binding and level-binding
constraints. Instead of a reduction of 66% from 71 to 25 in 2050 with a level target period of 2081-2100, the drop is
only 4.0% when the level target period is moved earlier by 40 years. However, a temporal distance of about 10 years



between the binding rate and level constraints is unlikely since the rate temperature increase will likely be gradually
reduced approaching the timing of the global temperature stabilization. In summary, the rate constraint becomes more
important for the combined metric as the temporal distance increases between the binding period of the rate constraint
and the level constraint.
**4    Discussion**
**4.1    Implementation**
A policy covering a range of species with a rate perspective can be implemented in various ways. Two/multi-basked
approaches have previous been discussed by Daniel et al. (2012); Fuglestvedt et al. (2000); Jackson (2009); Rypdal et
al. (2005). We present two approaches to combine the level and rate targets. The first method applies the level metric
and rate metrics individually, which we call the separate commitment approach. The second approach uses the
combined metric, thus, a common commitment.
**4.1.1 Separate commitments - two metrics**
Each party to an agreement now has two mitigation commitments, both quantified in terms of total $CO_2$-eq. emission
reductions, but using either the pure rate metric or the pure level metric to calculate the $CO_2$-eq. emissions. A dual
target is less flexible than a single target, which is likely more costly. Note that these commitments should be defined
(but could change) for all years following the time of the agreement. Figs. 3 and 4 show that the metric values varies
significantly with time of the emissions reductions, in contrast to the more traditional use of the (fixed) $GWP_{100}$.
**4.1.2 A common commitment - one metric**
In this case, each party has only one commitment in terms of $CO_2$-eq. emissions, using the combined rate and level
metric, with a chosen $\alpha$ ($0<\alpha<1$). As for the dual target case, the metric values varies with time, in particular for SLCFs,
with a sharp reduction for emissions after the end of the rate-binding period. The same argument holds also for this
case, i.e. that the parties need to know how the metric values change over time in order to implement cost-effective
policies. This case is probably simpler to implement for the parties as they have only one commitment to consider.
To implement cost-effective policies, the parties to the agreement need to know how the metric values change over
time when they plan investments that will reduce emissions for a longer period. The pulse metrics presented in this
paper, with their discontinuities, is only a building block. The abrupt change in emission metric value at the end of a
constraint period can make this emission metric confusing for decision makers. We discuss two options that will
remove this discontinuity. The first is to sum the absolute values of the rate metric and level metric, and then normalize
to $CO_2$:
$$M_{R\&L,i}(t_e) = \frac{AM_{R,i}(t_e)+AM_{L,i}(t_e)}{AM_{R,CO_2}(t_e)+AM_{L,CO_2}(t_e)}$$ (5)
This change of formula will smooth out the curve for the combined metric and give lower metric values during the
rate constraint period. Fig. S5 in Supporting Information is a remake of Figs. 3(A), 3(C), and 3(E) based on this
alternative formula. As we find our original formula simpler and easier to adjust the relative weighting, we prefer Eq.

301    (3).



The second alternative is to look at emissions over a longer period, which is often the case for decision makers leading
to emission changes lasting over a period. We give an illustrative example where decision makers can choose between
20 years of constant emissions of $CH_4$ versus $CO_2$. The emission metric value is then the average emission metric
value over a time period of 20 years following the implementation. The combined emission metric for $CH_4$ peaks in
2049 when considering individual years (see Fig. 3(A)), but an investment in 2049 has a metric value near the
minimum (see Fig. 5) due to the low metric values in the following years. The peak occurring around 2030 does not
have a discontinuity. The shape of the curve is similar for BC emissions, but with a relatively faster increase towards
2030 and larger decrease towards the end of the binding rate constraint period. Thus, a strategy for polluters and
policymakers cannot be based on the emission metric value for a single year, but a broader period relevant for
investments and policies.
**4.2   Cost-effective metrics**
The level and rate metrics presented here are purely physical based in that only physical quantities (like time, radiative
efficiency, etc.) are used to calculate their numerical values. However, from an economics point of view they represent
a cost-effective framework. On a fundamental level, emission metrics can either adopt a cost-effective or cost-benefit
approach which results in the metrics GCP (Manne and Richels, 2001) and Global Damage Potential (GDP) (Kandlikar,
1995), respectively (Tol et al., 2012). The GWP, which is also calculated from physical quantities only, can be derived
based on a cost-benefit approach where the benefit is optimized by weighting damages and costs. Alternatively, in the
cost-effective approach binding constraints are determined exogenously (e.g. the 1.5 and 2 $^{\circ}$C targets are based on
political negotiations), and policies are developed to reach the policy target in a cost-effective way. Under given
assumptions, it can be shown that the GTP metric is suited  for cost-effective approach to a level target (Tol et al.,
2012). The time horizon applied to calculate the metric values depends on the time interval when the constraints are
likely to be binding in an assumed baseline scenario.
**4.3   Weighting global emissions**
The impact in terms of $CO_2$-eq. emissions of different species depends on what perspective to take, whether level or
rate metric, what time horizon, or focusing on some other parameter. In Fig. 6(A), we show how different perspectives
compare based on the default level and rate metric cases, including GWP(100) and the Global Precipitation-change
Potential for pulse for a 20 year time horizon ($GPP_P(20)$) (Shine et al., 2015). For emissions in 2008 (EC, 2011;
Shindell et al., 2012), $CO_2$ is the most important contributor for all metrics, even the pure rate metric. If we keep the
emissions constant at the 2008 level and focus on the combined metric, the global BC and $CH_4$ emissions are given
little when they occur about 40-50 years before a binding constraint period (see Fig. 6(B)). The SLCFs increase their
influence closer to and during the rate-binding and level-binding constraint periods. The increase is most notable for
$CH_4$ in the first years, while the quick temperature response of BC leads to the largest increase for BC towards the end
of the rate constraint periods. However, global $CO_2$ emission is the most important or among the most important
drivers of temperature rates even during those binding periods. The most notable exception is the outsized influence
of BC in the final years of the rate and level binding constraint periods. These conclusions also hold for the rate metric
and level metric, as well as when applying emission scenarios such as the RCP6.0 (see Fig. S6 in the Supporting
Information).



### 4.4 Temperature development and RCPs

The time horizons of the level and rate targets are dependent on assumptions of the baseline scenario. In this study, we have presented applications of the level, rate, and combined metric with illustrative examples. However, the RCPs could be suitable for determining the timing of rate and level criteria. The global temperature increase and the decadal rate change according to RCP2.6, RCP4.5, RCP6.0, and RCP8.5, as well as historic data, is provided in Sect. 3 in the Supporting Information. According to RCP2.6, the global temperature stabilizes around 2060, while all other RCPs give an increase in temperature throughout the 21$^{st}$ century. Due to natural variability, the decadal temperature change can vary, historically cooling of almost -0.2 $^{0}$C/decade to warming up to 0.25 $^{0}$C/decade (Hansen et al., 2010; Morice et al., 2012; Smith et al., 2008). Similar fluctuations can be expected in the future. Current average decadal increase is approaching 0.2 $^{0}$C/decade. All RCPs indicate an anthropogenically driven rate of increase of about 0.2 $^{0}$C/decade in the next decades potentially giving the order of 0.4 $^{0}$C/decade when natural variability is added, which will likely be harmful for some of the plants and animals (Settele et al., 2014). A gradual reduction in the rate increase for the second half of the 21$^{st}$ century is seen for RCP2.6 and RCP4.5.

Our default case ends the binding rate constraint by 2050, which is partly consistent with a binding rate constraint of 0.2 $^{0}$C/decade in RCP4.5, or alternatively 0.1 $^{0}$C/decade in RCP2.6. The 1.5 or 2 $^{0}$C target from the Paris Agreement (UNFCCC, 2015) is for 2100. Collins et al. (2013) used the period 2081-2100 as a time proxy of climate change at the end of the 21$^{st}$ century, which is identical to our default level target period. Additional metric examples based on RCP4.5 and RCP8.5 with a range of different rate constraints are given in Sect. 8 in the Supporting Information.

For RCP8.5, the combined metric value increases throughout the century since the rate constraint is binding for the entire period or becomes binding (for >0.3 $^{0}$C/decade). On the other side, RCP4.5 gives a secondary maximum towards the end of the binding rate constraint period similar to Fig. 3 as the rate constraint is only binding in the first part of the century. In RCP2.6, one can argue that the level target is reached by 2060 as the global temperature increase since the pre-industrial time is set to fluctuate around 1.5 $^{0}$C from 2060 and for the rest of the century. A similar case is presented in Fig. 4(C).

Our default combined metric case, which is partly inspired by RCP4.5, has similar metric fluctuations as Manne and Richels (2001), with increasing (decreasing) metric values in the period of binding rate constraint for $CH_4$ ($N_2O$), and similar changes as the time of the level target is approached. Updated calculations by Ekholm et al. (2013) of the GCP for $CH_4$ for the 2 $^{0}$C level target and a combination of the 2 $^{0}$C level target combined with a rate constraint gave similar findings. However, the absolute values between our and their estimate differ, since they apply the GCP in some form based on economic assumptions and we apply analytical emission metrics.

The temperature development may be different from the baseline scenario, which could warrant the need for a regular updates of metric values. For instance, the temperature pathways are influenced by climate policy. If a stringent climate policy based on the metrics discussed here are applied, this will change this pathway as emissions are mitigated. This pathway change will further change the period when the rate constraint is binding, which changes the assumptions of the metric calculations i.e., a 'policy feedback.' If sustained and effective climate policy is practiced, the global





temperature trajectory will be lower than the baseline scenario. Not only is the temperature rate impacted, but also the
magnitude and the time horizon of the temperature stabilization or peaking. Another issue is what emission metric and
time horizon should be applied for emissions after both targets are reached, as continued climate policy is likely needed
to avoid further global warming, which potentially indicates that the emission metric values can be kept constant.
The rate metric presented here has a rate constraint that is either binding or not binding. Alternatively, a metric could
potentially be produced that assess different levels of temperature rates, for instance have a rate constraint starting at
the maximum allowable rate of change (e.g., 0.2 $^0$C/decade) that increases linearly in weight above that. This would
give additional weight to the periods with the largest temperature rates.
The anthropogenic contributed temperature rates may hypothetically fluctuate, hence, the rate constraint binding may
occur for several different periods separated by periods that are not binding. We have not given examples of this in
our analysis of a rate metric since such behavior is unlikely, but this behavior can easily be included. In Eq. (2), an
additional constraint period term between $t_{R3}$ and $t_{R4}$ can be added.
Some of the individual pathways of the RCPs indicate overshooting (Clarke et al., 2014), i.e. the level target is meet
in the long run but with a overshooting in the short-term. The AR5 WG3 Scenario database (Krey et al., 2014) shows
that overshooting pathways tend to have larger temperature rate increases, as well as for longer periods, than pathways
that approach level targets without overshoot (see Sect. 4 in the Supporting Information for details on scenarios
pathways that leads to $CO_2$ concentrations of 430-480 ppm and 530-580 ppm). Similar findings are previously
quantified by O'Neill and Oppenheimer (2004). Overshooting may lead to more ecological risks, as well as climate
feedback risks, than those pathways without overshoot (O'Neill and Oppenheimer, 2004). The overshot pathways
result in larger combined metric values for SLCFs than the other pathways with identical long-term level targets due
to longer rate constraint periods closer in time to the level constraint.
**5   Conclusion**
We have presented a physical and analytical rate metric concept that is compatible with the rate target described in
Article 2 in the UNFCCC. In addition, we have developed a combined metric that considers both the rate and level
target. We discussed and argued against alternative rate metrics derived from the rate of change. Several issues have
been discussed, such as determining when the rate and level constraints are binding and how to weight the rate and
level metrics. Further, we considered applying the rate of change perspective with two different approaches. One is to
argue for the long-term temperature stabilization target and the target of reducing the rate of climate change as two
different issues that need separate metrics. The other is to consider the two issues in one common framework that
warrant one combined metric with a selected weighting of the two targets. We presented some illustrative examples
of how these metrics can be used, as well as linking them to the RCPs. The suggested rate metric may be applied, as
the global temperature increase in the next decades can be harmful for some ecosystems. The total metric values for
SLCFs increase distinctly in periods when the rate constraint is binding, and the shorter the atmospheric perturbation
timescale is for a species. However, global emissions of $CO_2$ are the most important contributor when using the rate



metric, except for BC and partially $CH_4$ at the end of the binding period of rate constraint. The metrics presented here
for pulse emissions must be seen as building blocks for the users. The discontinuity in metric value at the end of a
constraint period can be difficult to communicate to users, while looking at emissions over a longer period resolves
this issue. We illustratively showed that an investment that leads to 20 years of sustained emissions gives a smoother
temporal metric profile for $CH_4$ than one based on pulses for each year. The utilization of these metrics are likely most
effective when the decision makers know how the metric values vary over time.
**Acknowledgements**
We thank Keith Shine and Drew Shindell for valuable comments. We thank Daniel Johansson for ideas on how to
combine rate and level metrics in one equation. The authors would like to acknowledge the funding by the Norwegian
Research Council Project "the Role of Short-Lived Climate Forcers in the Global Climate Regime."

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





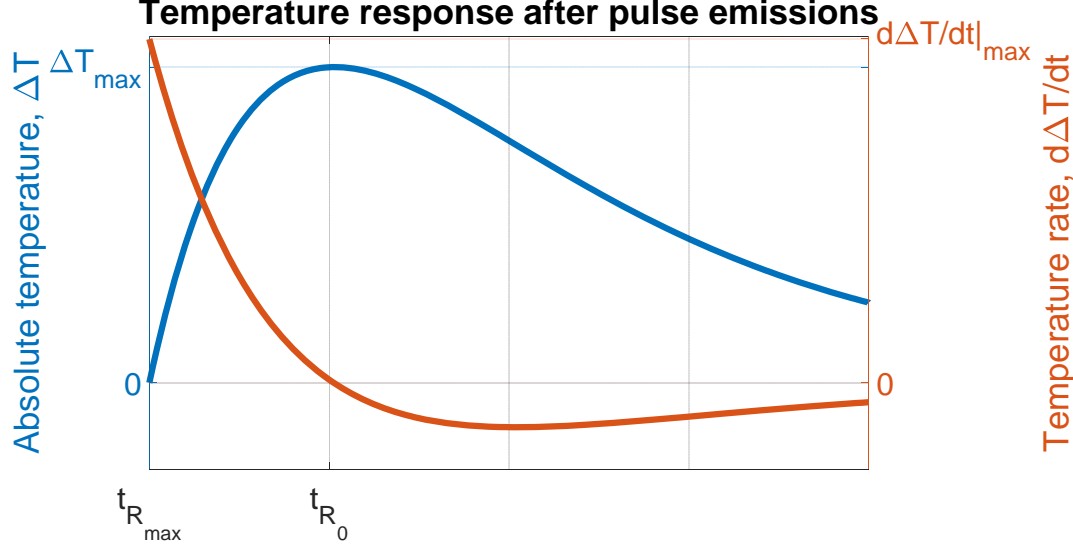


**Figure 1: A schematic of how the absolute temperature and temperature rate evolve after a pulse emission of a warming**

**species. The max rate of change occurs at $t_{Rmax}$ = 0. This figure is based on $CH_4$, but the principal is the same for all other**

**warming species.**




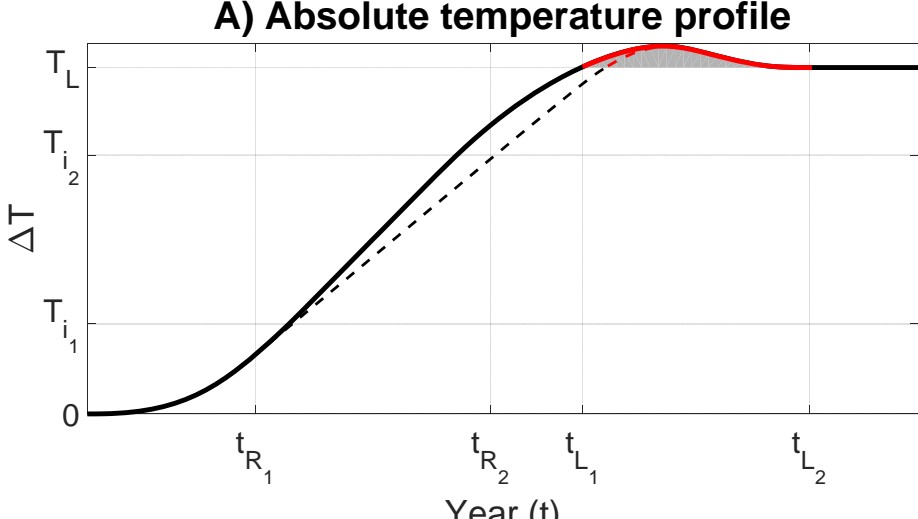


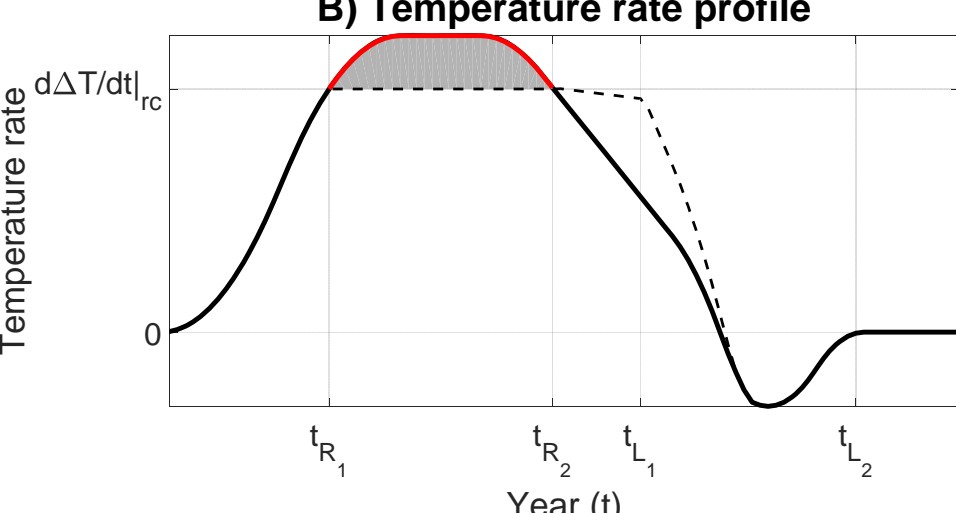


**Figure 2: A schematic of a baseline scenario where the global temperature initially have an increasing rate of change and eventually levels off. The total temperature change is given in A and the temperature rate in B. The level constraint above the level temperature $T_L$ occurs in the time period between $t_{L1}$ and $t_{L2}$ (shown in red). The rate of temperature increase is above some set critical level between $t_{R1}$ and $t_{R2}$ (shown in red) and makes the rate constraint binding for that period. The dotted line indicates an alternative baseline scenario that do not cross the rate constraint threshold, but leads eventually to the same total temperature increase.**








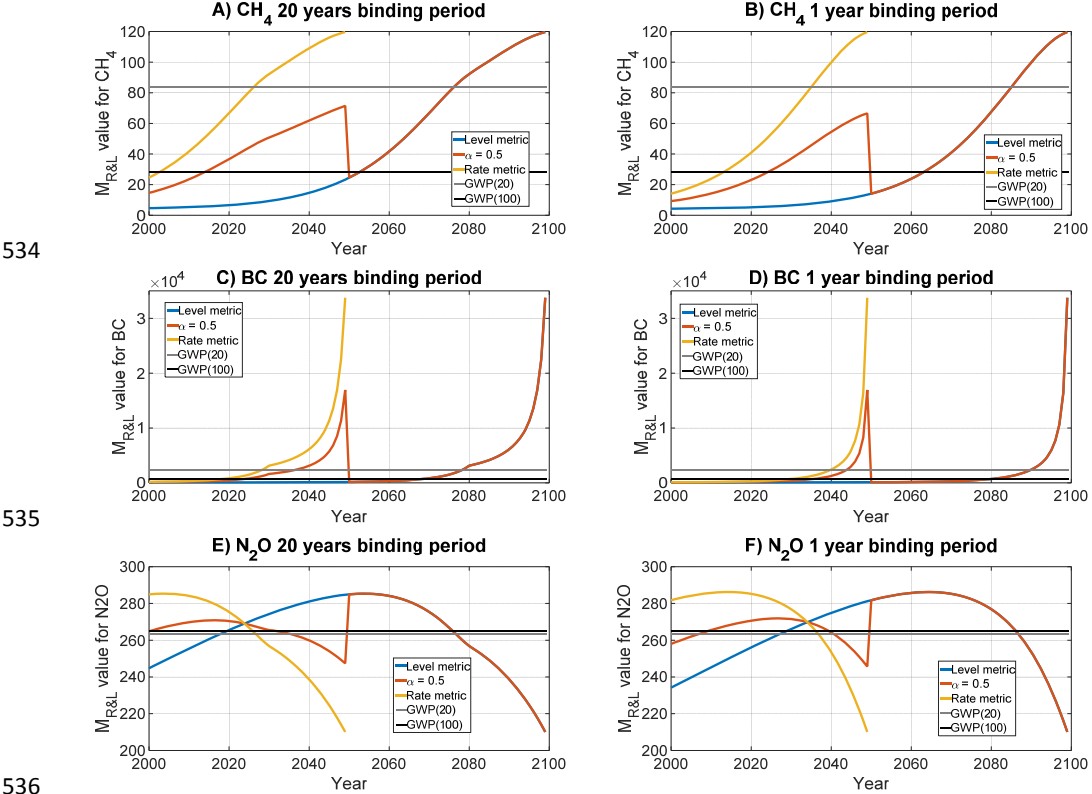

**Figure 3: Illustrative values for the level and rate metrics, as well as a combined metric based on equal weighting, for CH₄,**
**BC, and N₂O. Metric values for other weightings (α) are shown in Sect. 5 in the Supporting Information. The rate constraint**
**is binding for the period 2031-2050, and the level reached in 2081-2100 in A, C, and E. For B, D, and F, the rate is binding**
**in 2050 and level binding in 2100. α is the weight given to the rate metric in the period that is binding. GWP(20) and**
**GWP(100) values are given as reference.**





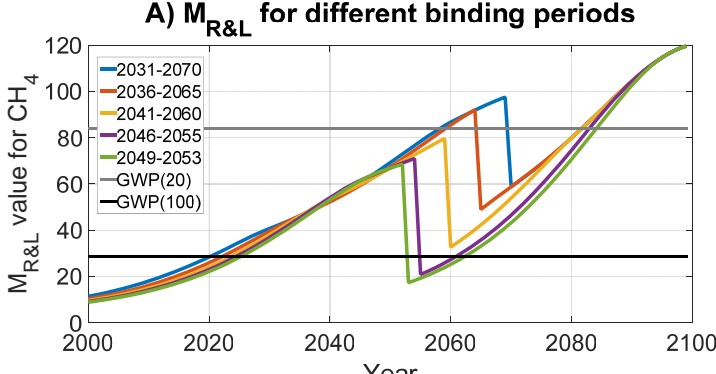


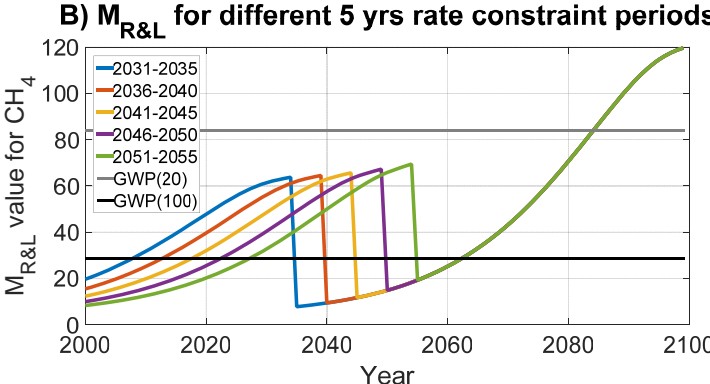


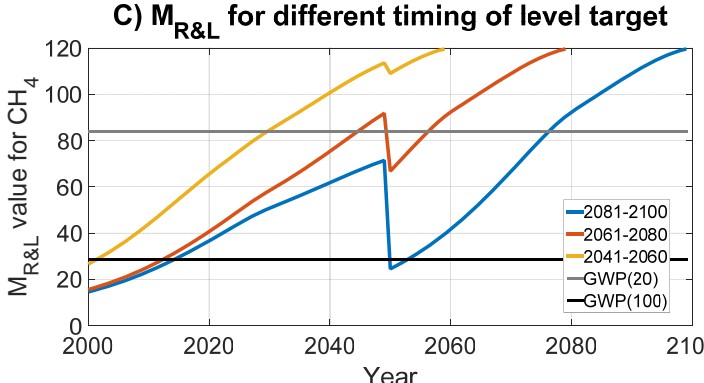


**Figure 4: How changing the timing of the level and rate targets influence the combined metric values. For all figures, the**
**length of the level constraint in years is identical to the length of the rate constraint period. All metric values are given**
**based on a weighting of α=0.5. In A, the length of the period of binding rate constraint (5-40 years) varies with midpoint in**
**2050. B shows a 5 years period of binding rate constraint at different times. C shows a sensitivity test of the timing of the**
**level target. GWP(20) and GWP(100) values are added to all figures as a reference.**






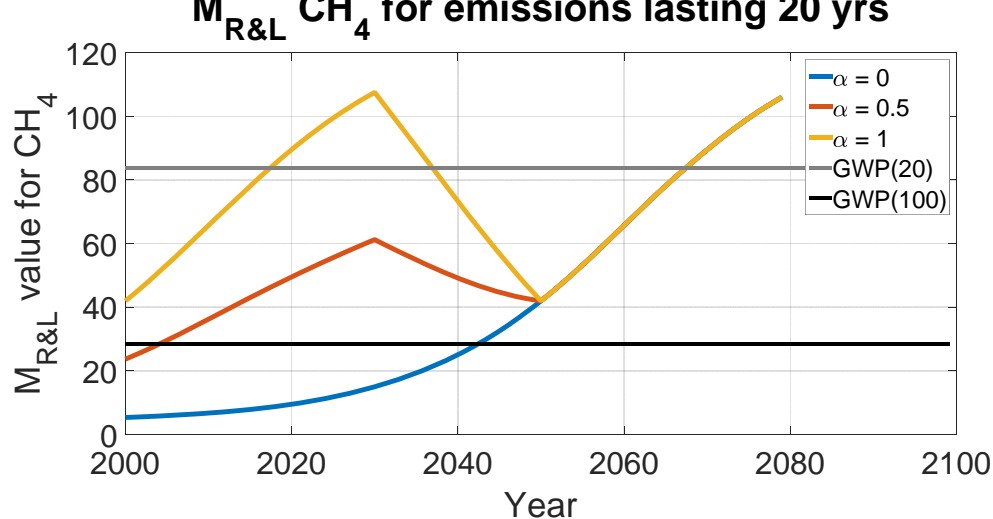


**Figure 5: Metric values for investments that lead to constant emissions over a period of 20 years based on the baseline scenario. The weighting is α=0.5. GWP(20) and GWP(100) values are given as reference. For the rate metric (α=1), we apply the level metric after the period of binding rate constraint.**






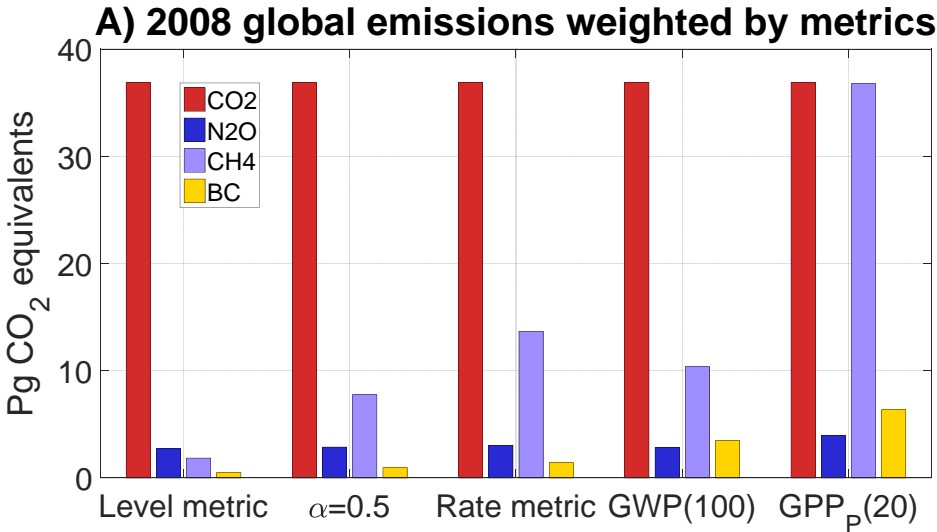


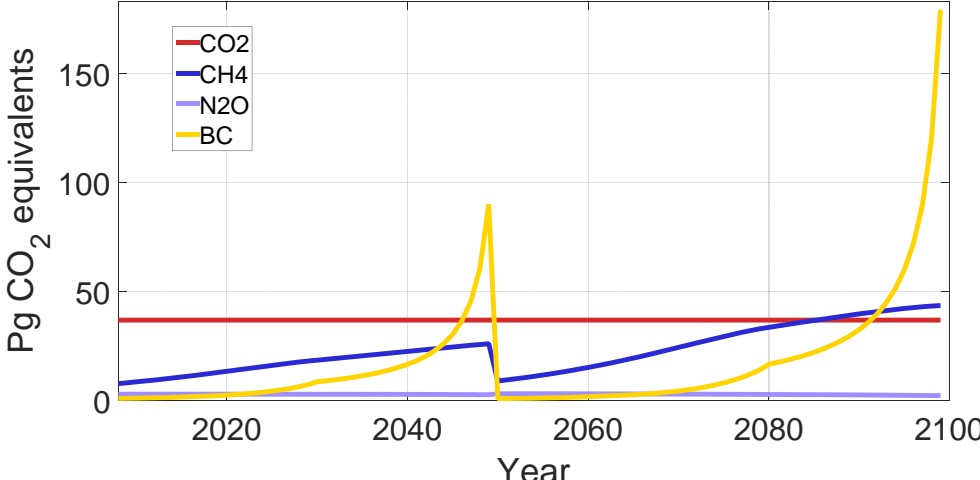


**Figure 6: A shows the global 2008 emissions weighted by different emission metrics. The calculations is based on Fig. 3,**
**with the rate constraint binding for the 2031-2050 period and level reached in 2081-2100. The level metric is here the same**
**as α=0 and the rate metric equal to α=1. These metrics are given equal weight with α=0.5, while GWP(100) and GPP$_P$(20)**
**are given for comparison. B is based on constant 2008 emissions for the rest of the century. The emissions are weighted with**
**the combined metric (α=0.5).**