# Peer review of "Combining temperature rate and level perspectives in emission 2 metrics"

_Earth System Dynamics, 2017_

## Referee Comment (RC1) · Anonymous Referee #1 · 17 Apr 2017

A review of "Combining temperature rate and level perspectives in emission metrics" by Aamaas et al. (Earth Syst. Dynam. Discuss., doi:10.5194/esd-2017-25, 2017)

The manuscript proposes a new metric that allows the comparison of different greenhouse gases' climate impacts against that of $CO_2$. The authors state that the proposed metric combines both temperature change and increase rate impacts of greenhouse gases, using linear weighting of the two components. There has been active research and discussion on climate metrics during past years, and new contributions on the topic might be useful. The manuscript is therefore interesting. The calculations seem to be executed well and the manuscript is pleasant to read.

However, my criticism focuses on the proposed metric itself.

1) There is a major fundamental issue in the proposed rate metric: it is not a rate metric.

[Figure]

Equation (2) doesn't integrate the temperature increase rate $R(t)$, but the temperature level $\Delta T(t)$. The proposed rate metric is therefore identical to the temperature level metric (equation 1), only that the integration limits are based on the years when the rate constraint is binding. It measures temperature level, not the rate, but on years in which the rate constraint is binding. This seems like a strange hybrid to my eye.

The authors mention in section 2 that integrating the increase rate equals the temperature change (i.e. AGTP in the metrics jargon). I believe this has led to the choice of integrating the temperature level and not the increase rate. This argumentation, however, doesn't change the fact that the metric doesn't measure the increase rate. The metric could be renamed, but this would make it less interesting: a variant of the GTP metric with integration over several years instead of a single-year endpoint.

2) There seems to be also a conceptual problem with the proposed metric. The integration ranges are defined to be the years where the chosen baseline scenario exceeds the chosen limit or rate constraints. Because the exceedance is a binary attribute, this definition can make the metric sensitive to the choices over baseline scenario or limits in some cases. The authors discuss this issue to some extent in sections 3.2 and 2.3, and figure 4 shows that the metric does vary considerably between different assumptions on when the constraint is binding. Yet, the general nature of the problem does not become evident from that discussion.

Imagine a scenario where temperature change is stabilized roughly at 2C, but with fluctuations around 2C because it's hard to hit the target spot-on in a dynamic system. This would render the absolute metric calculations of eq. (1) rather arbitrary. This example might be an irrelevant curiosity, but the binary nature of the metric can create problems for a wide number of scenario-limit combinations. Generally, the closer the scenario is on remaining below the limits or exceeding them, the more sensitive the metric will be to small changes in the scenario or the limits.

3) On the practical level, I would also anticipate that agreeing on the baseline scenario

would be a challenge, particularly given the sensitivity noted above. While this is not a flaw of the proposed metric in a scientific sense, it could severely limit its application in practice.

Based on the above arguments, I see the metric as a variant of the GTP with some added complications, which lead to possibly severe problems. It doesn't measure the temperature increase rate, as the label says. Due to the mis-labelling and design flaws, I don't see the metric or the manuscript to be of high quality, and regrettably have to suggest rejecting the manuscript.

Otherwise, there are a number of smaller issues on which the manuscript should be improved:

1) The manuscript resorts to inaccurate argumentation and lax rhetoric in some cases.

First, the authors justify the proposed metrics with Article 2 of the UNFCCC (rows 29-31, 61-62 and 67). I read the article carefully a few times, but I didn't find it mentioning temperature or rates in any way. I assume the authors have made their own interpretations on what the article means. There are no specific temperature goals in the article, unlike is stated on rows 29-31. On row 67 the authors state that the need for the metric is based on article 2. Yes, there might be a need, but it is not based on the article. The article only mentions the stabilization of greenhouse gas concentrations, which could be implemented in absence of climate metrics by setting separate concentration limits for each gas. (The article is mentioned again incorrectly at row 100, by stating that the rate causes damages. Yes is does, but article 2 doesn't state that.)

This argumentation gives a false impression that the main article of the UNFCCC would require a climate metric just like what is proposed in this article (row 397). I disagree strongly.

Second, the rate metric of equation (2) is motivated rather loosely on row 167 with "any additional warming is equally critical throughout the period of the binding rate

constraint". I don't agree with this statement, for marginal changes at higher levels or rates can inflict much higher damages. Also, if the statement were true, wouldn't it make sense to integrate also years other than those on which the limits are binding?

2) There are a number of points where the readability should be improved:

Row 14: Why 'baseline scenario'? Couldn't it be just 'scenario', as there is no alternative case to the 'baseline'?

Section 2: Mention explicitly that the metrics are not time-invariant with respect to the time of the emission, and this leads to that the metric is defined as a function of $t_e$ and $t$.

Rows 107 – 124: The notation (e.g. $AM$, $R_{max}$) is not explained.

Figure 2: Undefined expressions $T_{i1}$, $T_{i2}$ and $d\Delta T/dt|_{rc}$

Equations (1) to (3): $AM_x$ needs to be indexed with regard to $i$ (as is done in eq. 4)

---

## Referee Comment (RC2) · Anonymous Referee #2 · 21 Apr 2017

This paper proposes a new emission metric that combines rate and level targets. The rate of change perspective is important, and the conceptual framework is relatively well presented in the beginning. I have however two main issues with this work.

Firstly, I am not convinced on the value of this new metric in applications. While a rate metric is conceptually interesting and useful to be explored theoretically, the paper left an impression that overextends the applicability of the proposed new metric. This manuscript starts with Article 2 of the UNFCCC, however the interpretation is somewhat subjective, especially on the need for a metric compatible with the rate target. While previous literature suggested there might be some maximum acceptable temperature rate, right now it is not supported by as much evidence as the temperature level.  Such trend could certainly be critical for plants and animals if lasting for several decades, but the exact critical duration is also not clear, and additionally there is

natural climate variability which is not considered in the conceptual framework of this study. Some important assumptions are also made without much support (e.g., the baseline scenario). As such, the paper left with an impression that some groundwork has to be completed first for a robust rate metric to be applicable. At least the key assumptions in this paper should be clearly listed and better defended. The authors also admitted that the political feasibility might be low.

Secondly, as a pure conceptual work, the framework is not described clearly in this paper, especially for ESD's diverse readership. While figure 1 and 2 are still relatively easy to follow without explaining each symbol, the major part of the writing contains numerous distracting jargons that cumulatively impede understanding of the work. Section 2 starts with Alternative rate metrics without specifying alternative to what (to GTP?). If the focus is on improvement to GTP metric, then the GTP metric itself should at least be introduced and highlight the modifications in this new metric. The paper also tries to combine the rate aspect and the $CO_2$-eq aspect, which also dilutes the focus.

In summary, my recommendations for the authors are: 1) frame this paper differently without overextending too much on the applicability; 2) Either making the symbols and paper organization clear to follow, or submitting to a more specific journal. As the suggestions require a complete rework, unfortunately I cannot recommend publication of the paper in ESD.
* * *

---

## Author Comment (AC1) · 30 May 2017

We thank the reviewers for comments. Our responses are given below in red. We first give a general comment addressing some of the main issues in both reviews, before we reply to the two reviews in detail individually.

While we agree with the reviewers that our manuscript should be clearer in the argumentation and will benefit from this round of revisions, we disagree with the reviewers on several of the key issues raised. We hope the Editor sees the importance of discussing the framework around metrics on temperature rate perspectives, and that a revised version of this manuscript could do that.

We reject the idea that a rate metric cannot be linked to Article 2 of UNFCCC. But we will improve the formulations to better show the linkage. We are not the first to make such a connection. Manne & Richels (2001) explicitly discuss UNFCCC Article 2 and say that concern such be on two indicators, temperature change and rate of temperature change. They propose trade-off ratios based on these two concerns, just as we do. Opphenheimer and Petsonk (2005) discussed the historical origins and interpretations of Article 2. They are clear on that "dangerous anthropogenic interference" is by many translated in to a level of global temperature and "within a time frame" understood by slowing the rate of temperature change. We will add reference to Opphenheimer and Petsonk (2005) to strengthen the perceived link between Article 2, temperature targets, and thus metrics suitable for these targets. We will revise parts of the manuscript that is on Article 2, so it becomes clearer that Article 2 does not say we need level and rate metrics but common understanding of Article 2 leads to these metrics. Here and in the following "rate metric" is understood as a metric that is intended to support policies to slow the rate of warming during a period of time when this is believed to cause damage. We will also rewrite and add material to Section 4.4 and 5 to discuss our rate metric is linked to previous and new literature.

A discussion of near term (such as in a rate perspective) and long term (such as in a level perspective) targets have been discussed since the First Assessment Report from IPCC. This is also an on-going discussion, such as by reading the Science edition published on May 5th 2017. Two articles were discussing these near-term and long-term issues and how emission metrics should be used (Ocko et al., 2017; Shindell et al., 2017). Ocko et al. (2017) argue for two different timescales to reflect both the short-term and long-term, which our manuscript was a try to widen the conceptual aspects of. Shindell et al. (2017) push the importance of a near-term goal to try to reduce the pace of climate change. They propose a mean AGTP with a time horizon of 25 years, which in practice is a special case of our rate metric proposed in the manuscript. This equals to a rate binding from today and 25 years onwards. The Climate & Clean Air Coalition (CCAC) has discussed this specific metric, and, hence, there is political interest in the perspectives discussed in our manuscript (for more details, see the discussion on mean AGTP over 25 years in the CCAC here: http://enb.iisd.org/climate/ccac/wgspd20/html/enbplus172num35e.html). As Science is publishing commentaries on this, we hope that Earth System Dynamics will publish our manuscript that, in our mind, is providing an academic framework for these issues.

Both reviewers seem to believe by establishing the framework we are advocating for the metric with a rate perspective to be used in mitigation policies. However, our objective is more academic, acknowledging the need to establish this framework should policymakers call for a multi-component mitigation policy to slow down the near-term warming. Given the recent commentaries and the discussions within CCAC, there is clearly a need for such a framework.

Oppenheimer, M., and Petsonk, A.: Article 2 of the UNFCCC: Historical origins, recent interpretations, Climatic Change, 73, 195--226, 2005.

Ocko, I. B., Hamburg, S. P., Jacob, D. J., Keith, D. W., Keohane, N. O., Oppenheimer, M., Roy-Mayhew, J. D., Schrag, D. P., and Pacala, S. W.: Unmask temporal trade-offs in climate policy debates, Science, 356, 492-493, 10.1126/science.aaj2350, 2017.

Shindell, D., Borgford-Parnell, N., Brauer, M., Haines, A., Kuylenstierna, J. C. I., Leonard, S. A., Ramanathan, V., Ravishankara, A., Amann, M., and Srivastava, L.: A climate policy pathway for near- and long-term benefits, Science, 356, 493-494, 10.1126/science.aak9521, 2017.

**Anonymous Referee #1**

The manuscript proposes a new metric that allows the comparison of different greenhouse gases' climate impacts against that of CO2. The authors state that the proposed metric combines both temperature change and increase rate impacts of greenhouse gases, using linear weighting of the two components. There has been active research and discussion on climate metrics during past years, and new contributions on the topic might be useful. The manuscript is therefore interesting. The calculations seem to be executed well and the manuscript is pleasant to read.

However, my criticism focuses on the proposed metric itself.

1) There is a major fundamental issue in the proposed rate metric: it is not a rate metric. Equation (2) doesn't integrate the temperature increase rate R(t), but the temperature level T(t). The proposed rate metric is therefore identical to the temperature level metric (equation 1), only that the integration limits are based on the years when the rate constraint is binding. It measures temperature level, not the rate, but on years in which the rate constraint is binding. This seems like a strange hybrid to my eye.

   The authors mention in section 2 that integrating the increase rate equals the temperature change (i.e. AGTP in the metrics jargon). I believe this has led to the choice of integrating the temperature level and not the increase rate. This argumentation, however, doesn't change the fact that the metric doesn't measure the increase rate. The metric could be renamed, but this would make it less interesting: a variant of the GTP metric with integration over several years instead of a single-year endpoint.

The reviewer makes a good point, that a rate metric should measure the rate! We did discuss this (option 1 in Section 2), but found that using the instantaneous rate has several issues. We see now, that we did not clearly outline these issues and why we therefore considered options 2 and 3 in Section 2. It should be said, that options 2 and 3 are essentially the average rate of change over a time period. However, we see now that we removed the time dimension for the time period of integration from metrics (we did this as in a normalized metric, the time dimension will cancel in the numerator and denominator). We see that it would be beneficial to rewrite Section 2, and more clearly outline why we did not use the instantaneous rate and instead focus on the average rate, which to a constant, is the same as the level over a given period.

It is worth noting that some emission metrics don't necessarily relate directly to a physical interpretation. The GWP(100) is a good case in point. The GWP represents the integrated forcing, which does not correlate to the temperature increase (Shine et al., 2005). However, we still apply the GWP(100) as a decent measure of the long-term temperature increase. In a similar manner, we believe that using the contribution to temperature increase over some period (proportional to the average rate) provides the best basis for a metric regarding how individual pulse emissions contribute to the rate of warming in that period.

While writing this manuscript, we discussed among ourselves the obvious choice of making a metric based on rate of temperature change. A challenge with the instantaneous rate is the decrease seen in the temperature rate seen shortly after emissions of SLCFs, which might lead to perverse incentives to increase SLCF emissions as a measure to reduce the rate. We see that the thought process we had, and not just the conclusion, would be interesting for the paper. We will therefore rewrite Section 2 to clearer state why we are not proposing metrics based on temperature rate directly. We tried to be short in words, but will expand in the revisions. Section 2 will be rearranged into two subsection, one describing metric based on instantaneous rate and one on alternative rates that equals different types of AGTPs.

Shine, K. P., Fuglestvedt, J. S., Hailemariam, K., and Stuber, N.: Alternatives to the Global Warming Potential for Comparing Climate Impacts of Emissions of Greenhouse Gases, Climatic Change, 68, 281-302, 10.1007/s10584-005-1146-9, 2005.

2) There seems to be also a conceptual problem with the proposed metric. The integration ranges are defined to be the years where the chosen baseline scenario exceeds the chosen limit or rate constraints. Because the exceedance is a binary attribute, this definition can make the metric sensitive to the choices over baseline scenario or limits in some cases. The authors discuss this issue to some extent in sections 3.2 and 2.3, and figure 4 shows that the metric does vary considerably between different assumptions on when the constraint is binding. Yet, the general nature of the problem does not become evident from that discussion.

Imagine a scenario where temperature change is stabilized roughly at 2C, but with fluctuations around 2C because it's hard to hit the target spot-on in a dynamic system. This would render the absolute metric calculations of eq. (1) rather arbitrary. This example might be an irrelevant curiosity, but the binary nature of the metric can create problems for a wide number of scenario-limit combinations. Generally, the closer the scenario is on remaining below the limits or exceeding them, the more sensitive the metric will be to small changes in the scenario or the limits.

This issue of scenario, baselines, and time horizons raised by the reviewer is relevant for all metrics, not just the metrics discussed in the manuscript. The GWP(100) also has a baseline problem, most just don't know about it! Several papers have discussed how selecting scenarios and baselines will impact the metric value, such as Reisinger et al. (2011). Yes, we agree this is a problem, but it is a problem that we have to deal with regardless of the metric.

These issues are addressed somewhat in the manuscript, as mentioned by the reviewer. In the seventh paragraph of Section 4.4, we briefly discuss an alternative that is not binary, as more weight can be given with increasing violation of the constraint. We argue in the next paragraph that a fluctuating temperature curve due to anthropogenic forcing is unlikely, and, hence, the variability that the reviewer

writes about is not very likely. In Section 2.3, we state that we focus on the impact of anthropogenic forcing, not on natural fluctuations. If we were to use temperature observations or output from general circulation models, this variability could be an issue. As we look at the anthropogenic forcing, this is much less of a problem. This manuscript is an attempt to give the framework of how rate and level metrics can be applied. The focus is on the big picture.

Reisinger, A., Meinshausen, M., and Manning, M.: Future changes in global warming potentials under representative concentration pathways, Environmental Research Letters, 6, 024020, 10.1088/1748-9326/6/2/024020, 2011.

3) On the practical level, I would also anticipate that agreeing on the baseline scenario would be a challenge, particularly given the sensitivity noted above. While this is not a flaw of the proposed metric in a scientific sense, it could severely limit its application in practice.

We do not disagree with the reviewer. In the manuscript, we do not advocate for a certain baseline scenario, but we test how this metric concept works with different starting points. The main point with this article is not to recommend certain metric values, but to explore scientifically the rate and level perspectives. Even though these ideas might be difficult to implement, we believe that it is of interest to explore these concepts scientifically. Further work may show that the metrics we present in our manuscript are of limited use. As we mentioned in the previous point, the baseline issue is not unique to our metric, and this will be outlined further in the revised manuscript.

Based on the above arguments, I see the metric as a variant of the GTP with some added complications, which lead to possibly severe problems. It doesn't measure the temperature increase rate, as the label says. Due to the mis-labelling and design flaws, I don't see the metric or the manuscript to be of high quality, and regrettably have to suggest rejecting the manuscript.

We believe that we have not sufficiently outlined the context of the paper. Referring back to the Science papers mentioned earlier, they essentially recommend using two metrics, one with a short and a long-term horizon, to essentially represent rate and level metrics. Indeed, the IPCC First Assessment Report linked a few decades (20 year time horizon) to near-term climate change and the rate of change and 100 years or more to the level or to cumulative change (like see level). Most of all, we see our paper as outlining the conceptual reasoning behind this. The short-term rate, is more the average rate over a period, not the instantaneous rate. Further, we don't intend to advocate using the combined rate and level metric, but we intend to outline how it may look if one took that option.

Otherwise, there are a number of smaller issues on which the manuscript should be improved:

1) The manuscript resorts to inaccurate argumentation and lax rhetoric in some cases.

First, the authors justify the proposed metrics with Article 2 of the UNFCCC (rows 29-31, 61-62 and 67). I read the article carefully a few times, but I didn't find it mentioning temperature or rates in any way. I assume the authors have made their own interpretations on what the article means. There are no specific temperature goals in the article, unlike is stated on rows 29-31. On row 67 the authors state that the need for the metric is based on article 2. Yes, there might be a need, but it is not based on the article. The article only mentions the stabilization of greenhouse gas concentrations, which could be implemented in absence of climate metrics by setting separate concentration limits for each gas. (The

article is mentioned again incorrectly at row 100, by stating that the rate causes damages. Yes is does, but article 2 doesn't state that.)

See our general comment where we state the clear linkage between Article 2 and metrics. We agree that the argumentation between Article 2 and the metrics presented could have been better. We will improve the introduction section and all parts of the manuscript that makes the connection with Article 2 and the metrics. As written in the general comment, we will refer to Opphenheimer and Petsonk (2005) and related literature on this.

We agree that Article 2 only mention greenhouse gases, but we would like to point out that does not exclude short-living greenhouse gases. We also believe that this formulation should be thought of as broader, as the global warming impact of black carbon was put in the spotlight several years later in the early 2000s, such as with Menon et al. (2002) on climate effects of black carbon aerosols in China and India.

We will deal with all the minor changes on specific rows as indicated by the reviewer.

This argumentation gives a false impression that the main article of the UNFCCC would require a climate metric just like what is proposed in this article (row 397). I disagree strongly.

See our general comment above. We will change the formulation to say that the UNFCC text leads to emission metrics.

Interest in this is for instance seen from CCAC, which proposes a metric for the near term temperature is in practice is very similar to the rate metric proposed in our submission. They proposed mean AGTP over 25 years, http://enb.iisd.org/climate/ccac/wgspd20/html/enbplus172num35e.html

Second, the rate metric of equation (2) is motivated rather loosely on row 167 with "any additional warming is equally critical throughout the period of the binding rate constraint". I don't agree with this statement, for marginal changes at higher levels or rates can inflict much higher damages. Also, if the statement were true, wouldn't it make sense to integrate also years other than those on which the limits are binding?

The sixth paragraph in Section 4.4 of our manuscript discusses the possibility to weigh certain periods more than others. Our proposed metric is meant to be simplified, while we agree with the reviewer that more details included could improve the metric values. We do not agree with the last comment. Part of our argument is that small temperature rates can be tolerated as ecosystems can naturally adapt to those changes. In periods with temperature rates below the rate constraint, we do not agree with the reviewer that those years should also be counted.

2) There are a number of points where the readability should be improved:

Row 14: Why 'baseline scenario'? Couldn't it be just 'scenario', as there is no alternative case to the 'baseline'?

We would like to keep the wording "baseline scenario" as the emission metric values for level and rate perspectives have to be calculated on something that can see as a starting point, the most likely scenario, a baseline scenario. By applying these metrics in policies, this might lead to cost-effective emission reductions relative to the expect baseline. This can lead to alternative pathways.

Section 2: Mention explicitly that the metrics are not time-invariant with respect to the time of the emission, and this leads to that the metric is defined as a function of te and t.

We have added this sentence to Section 2:

"The proposed emission metrics are not time-invariant with respect to the time of the emissions, and, hence, the emission metrics are dependent on emission time $t_e$."

Rows 107 – 124: The notation (e.g. AM, Rmax) is not explained.

Notation is explicitly explained.

Figure 2: Undefined expressions Ti1, Ti2 and dT=dtjrc

They are now expressed.

Equations (1) to (3): AMx needs to be indexed with regard to i (as is done in eq. 4)

All equations are indexed in regards to species i.

**Anonymous Referee #2**

This paper proposes a new emission metric that combines rate and level targets. The rate of change perspective is important, and the conceptual framework is relatively well presented in the beginning. I have however two main issues with this work.

Firstly, I am not convinced on the value of this new metric in applications. While a rate metric is conceptually interesting and useful to be explored theoretically, the paper left an impression that overextends the applicability of the proposed new metric. This manuscript starts with Article 2 of the UNFCCC, however the interpretation is somewhat subjective, especially on the need for a metric compatible with the rate target. While previous literature suggested there might be some maximum acceptable temperature rate, right now it is not supported by as much evidence as the temperature level.âˇA ´l Such trend could certainly be critical for plants and animals if lasting for several decades, but the exact critical duration is also not clear, and additionally there is natural climate variability which is not considered in the conceptual framework of this study. Some important assumptions are also made without much support (e.g., the baseline scenario). As such, the paper left with an impression that some groundwork has to be completed first for a robust rate metric to be applicable. At least the key assumptions in this paper should be clearly listed and better defended. The authors also admitted that the political feasibility might be low.

See our general comment above. We will change the text, especially in the introduction section, to better show how Article 2 can lead to the metrics presented in our manuscript (see the general comment and response to Reviewer 1). The relationship between Article 2 and the rate metric is justified with literature (Oppenheimer & Petsonk, 2005).

We agree with the reviewer that uncertainties exist, especially damage functions on climate change, but we think that should not stop scientific studies to investigate potential frameworks. Our study is meant to be an exploration, not a proposition to use that and this metric value. The numbers used to show how the metrics will work like are based on semi arbitrary examples, such as what baseline scenario to use.

We are not necessary advocating for this metric, but we see a discussion is needed on this given past and potential future policy interest. The manuscript was written due to academic curiosity. The rate metric issue has already been discussed by policymakers, and we think academic investigation of this topic is important. Temperature increases in the short-term, and, thus, the temperature rate, is discussed in the science-policy interface in the CCAC. The CCAC has proposed a metric that in practice is very similar to the rate metric proposed in our submission (as stated in the general comment): http://enb.iisd.org/climate/ccac/wgspd20/html/enbplus172num35e.html. While the political feasibility may be low, that is not a reason not to show academic interest in such issues.

Yes, we agree that it is highly uncertain what can be seen as an acceptable rate warming. But Oppenheimer and Petsonk (2005) show in the discussion of how Article 2 is related to rate warming targets that rate targets have been proposed as early as in 1988. The proposed target was 0.1 °C per decade, similar to rate levels discussed in our manuscript. Due to the historic and current policy interest of this issue, we think our study is needed.

Secondly, as a pure conceptual work, the framework is not described clearly in this paper, especially for ESD's diverse readership. While figure 1 and 2 are still relatively easy to follow without explaining each symbol, the major part of the writing contains numerous distracting jargons that cumulatively impede understanding of the work. Section 2 starts with Alternative rate metrics without specifying alternative to what (to GTP?). If the focus is on improvement to GTP metric, then the GTP metric itself should at least be introduced and highlight the modifications in this new metric. The paper also tries to combine the rate aspect and the CO2-eq aspect, which also dilutes the focus.

We think ESD is a suitable journal for our manuscript as ESD has published several papers on emission metrics and has a broad scope. Our intentions of this paper is to shed light on the framework needed if somebody were to use rate and level metrics. The numbers in themselves are not important.

To help the readers not too familiar with emission metrics, we have added definitions of AGTP and GTP in the fourth paragraph of Section 1. Section 2 will be rewritten and the alternative metrics will be discussed in larger detail, which should answer most of the reviewer's concern.

All metrics can be used to calculate CO2-eq. GWP(100) is often assumed to give CO2-eq., but all metrics can be used the same way. We think it helps to calculate CO2-eq. with different metrics to show how importance the different aspects are.

In summary, my recommendations for the authors are: 1) frame this paper differently without overextending too much on the applicability; 2) Either making the symbols and paper organization clear to follow, or submitting to a more specific journal. As the suggestions require a complete rework, unfortunately I cannot recommend publication of the paper in ESD.